# Survival of Frail Elderly with Delirium

**DOI:** 10.3390/ijerph19042247

**Published:** 2022-02-16

**Authors:** Guillermo Cano-Escalera, Manuel Graña, Jon Irazusta, Idoia Labayen, Ariadna Besga

**Affiliations:** 1Department of Computer Science and Artificial Intelligence, University of the Basque Country (UPV/EHU), Paseo Manuel de Lardizabal, 1, 20018 Donostia-San Sebastian, Spain; manuel.grana@ehu.es; 2Computational Intelligence Group, University of the Basque Country (UPV/EHU), 20018 Donostia-San Sebastian, Spain; 3Department of Physiology, Faculty of Medicine and Nursing, University of the Basque Country (UPV/EHU), 48013 Bilbao, Spain; jon.irazusta@ehu.eus; 4BioCruces Health Research Institute, 48903 Barakaldo, Spain; 5Institute for Innovation & Sustainable Development in Food Chain (IS-FOOD), Public University of Navarra, 31006 Pamplona, Spain; idoia.labayen@unavarra.es; 6BioAraba, Health Research Institute, Hospital Universitario de Araba, Department of Medicine, 01004 Vitoria, Spain; ariadna.besgabasterra@osakidetza.eus; 7Biomedical Research Centre in Mental Health Network (CIBERSAM) G10, Spain

**Keywords:** delirium, frailty, survival, polypharmacy, ageing population, hospital admission

## Abstract

This study aims to determine when frailty increases the risks of delirium mortality. Hospital patients falling into the elderly frail or pre-frail category were recruited, some without delirium, some with delirium at admission, and some who developed delirium during admission. We screened for frailty, cognitive status, and co-morbidities whenever possible and extracted drug information and mortality data from electronic health records. Kaplan–Meier estimates of survival probability functions were computed at four times, comparing delirium versus non delirium patients. Differences in survival were assessed by a log-rank test. Independent Cox’s regression was carried out to identify significant hazard risks (HR) at 1 month, 6 months, 1 year, and 2 years. Delirium predicted mortality (log-rank test, *p* < 0.0001) at all four censoring points. Variables with significant HRs were frailty indicators, comorbidities, polypharmacy, and the use of specific drugs. For the delirium cohort, variables with the most significant 2-year hazard risks (HR(95%CI)) were: male gender (0.43 20 (0.26,0.69)), weight loss (0.45 (0.26,0.74)), sit and stand up test (0.67 (0.49,0.92)), readmission within 30 days of discharge (0.50 (0.30,0.80)), cerebrovascular disease (0.45 (0.27,0.76)), head trauma (0.54 22 (0.29,0.98)), number of prescribed drugs (1.10 (1.03,1.18)), and the use of diuretics (0.57 (0.34,0.96)). These results suggest that polypharmacy and the use of diuretics increase mortality in frail elderly patients with delirium.

## 1. Introduction

Delirium is a severe psychiatric syndrome characterized by a sharp change in attention, awareness, and the cognitive state of the patient [1,2,3] that is highly prevalent in hospitalized older patients [4], and it is a common complication after surgery [5,6,7]. In older people, the most frequent presentation of delirium is hypoactive [8], showing psychomotor slow down, bradypsychia, slow language, apathy, and inhibition. Most frequent precipitating factors in community-dwelling older individuals are infectious diseases, followed by drugs and hydro-electrolytic disorders [9]. Machine learning tools have been proposed for the identification of delirium onset risk factors [10,11]. The duration of delirium is variable, persisting for weeks or months in 20% of patients with dementia [12], which is a high risk factor for delirium in older patients [13], while delirium onset often leads to cognitive deterioration and dementia [3,14,15]. There is a strong relation between frailty [16,17,18,19] and delirium in older patients [20], with frailty identified as a predictor of delirium [21,22] and delirium onset as a strong biomarker for underlying frailty in older adults [23].

Delirium prevalence is highly variable depending on the study, population, and environmental conditions [24,25,26]. Delirium is the second most prevalent psychiatric syndrome in the hospital environment [27]. In the older population, delirium carries higher mortality risks, longer hospital stays, and impedes cognitive and functional capabilities [28]. An estimated 20% of hospitalized patients over 65 years in the US suffer complications due to delirium [29]. A recent study carried out over three emergency departments found that the prevalence of delirium was 19.1% [30], while a recent proposal a for review and meta-analysis raises the estimate to 40% of emergency department admittances [31].

Contrary to other medical conditions, delirium-associated mortality has remained unchanged in recent decades [32]. The onset of delirium increases the long-term mortality risk after intensive care unit (ICU) survival [33,34,35,36,37] and hospital discharge [38], while the core symptoms of delirium increase the mortality risk for older people by themselves [39]. For hospitalized older patients, an early diagnosis and treatment are desirable in order to avoid a severe worsening of the condition [40]. The association of frailty and delirium does increase the mortality risk during critical care in intensive care units (ICU) [35], on top of frailty being a strong predictor of mortality across the life course [41].

This paper presents a survival analysis of older and mostly frail patients admitted to the University Hospital of Alava (UHA), Vitoria, Spain, that were diagnosed with delirium either at admission (prevalent cohort) or while staying in the hospital (incident cohort). We compare the survival probabilities of the non-delirious patients versus delirious patients at four censoring times: 1 month, 6 months, 1 year, and 2 years. The first objective of the study is to assess if delirium predicts worse survival probabilities. The second objective of the study is to identify the variables showing greater hazard risks (HRs) at the considered censoring times.

## 2. Materials and Methods

### 2.1. Study Design and Subjects

Figure 1 describes the patient recruitment process. The patients were admitted to the services of internal medicine and neurology at the UHA. Delirium was evaluated by the confusion assessment method (CAM) [42] at admittance and during the hospital stay. Partial noninvasive frailty and cognitive assessment was carried out also at admittance. Recruitment and tests requiring physical and cognitive involvement of the patient were carried out if and when the patient was stabilized and able. Patient recruitment was implemented in the period from September 2017 to September 2018. Mortality follow-up until January 2021 was performed by querying the institutional EHR. Following open science practices, the anonymized dataset was preliminary published in the Zenodo public repository [43].

The inclusion criteria were the following: age above 70 years, scoring more than 20 in the Spanish version of the Mini Mental State Questionnaire (MMS), ability to walk with or without aids and to perform simple physical tests, able to understand and follow simple instruction, and being able to sign the informed consent at the time of recruitment. Exclusion criteria were the following: history of chronic kidney disease, had suffered a heart attack in the last 3 months, have been unable to walk, have suffered any fracture of the upper or lower limbs in the last 3 months, have been suffering from severe dementia, a history of autoimmune neuromuscular disorders (for example, myasthenia gravis, Guillain–Barré syndrome, inflammatory myopathies) or amyotrophic lateral sclerosis. Furthermore, when the patients were moved into a different medical service unit of the hospital, or refused to sign the informed consent. Recruitment achieved an almost gender-balanced cohort of 741 patients. This paper reports separate analyses for patients diagnosed with delirium at admission (N = 170, *prevalent* cohort), and all the patients that suffered delirium at some point of the hospital stay (N = 200, *incident* cohort, includes *prevalent* cohort).

To assess the functional status of the patient, the following tests were applied by experienced researchers (AB, JI): Short Physical Performance Battery (SPPB), Fried’s frailty index (FFI), and Barthel’s index score (BIS) measuring performance in activities of daily living. The SPPB test [44] includes 3 tests: balance test, walking speed over 4 meters, and sitting and stand up five times. FFI test [17] defines frailty phenotype characterized by involuntary weight loss, fatigue, muscular weakness, slow march, and decay of physical activity. Barthel’s scale [45] measures the physical handicap. The nutritional state was assessed using the Mini Nutritional Assessment—Short Form (MNA-SF)—as a mean to identify older subjects at risk of malnutrition before the apparition of severe changes in weight or serum or protein concentrations [46,47]. The Pfeiffer’s Brief Screening Test for Dementia (PBSTD) [48,49] was applied for mental condition assessment. Additionally, we considered the number of falls in the previous month. All in all, the study covered more than 300 variables. Experienced members of the research team (AB, GC) accessed the electronic health records (EHR) to extract sociodemographic data, survival data, clinical data such as comorbidities, and pharmacological information.

Missing values were corrected by setting the default values after consultation with clinicians and double checking on the EHR. After data curation, variables with more than 20% cases with missing values were ignored to ensue processing. Aggregated variables, such as FFI outcomes, were included in the hazard risk analysis, along with their component variables.

### 2.2. Statistical Methods

Data were processed in Rstudio 1.2 and R 3.6.3 (www.r-project.org, accessed on 1 February 2022) using packages HSAUR2, Survival, and Survminer. The Kaplan–Meier estimate of the survival function and its variance [50,51,52] was computed at four censoring dates, namely: 1 month, 6 months, 1 year, and 2 years. Log-rank test [52] assessed the statistical significant difference of the probability of survival curves of delirium versus non-delirium patients. Significance threshold was 0.05. Separate analyses for the prevalent and the incident cohorts were carried out. Cox’s regression [52,53] was computed to assess the hazard risks (HR) of the variables at each of the four censoring times. Variables were split into families as follows: (a) demographics, including frailty tests, and number of drugs; (b) comorbidities; (c) pharmacological variables. Independent multivariate Cox’s regressions were carried out over each family. Proportional hazard assumption was tested for all Cox regression analyses by Schoenfeld residuals, and p>0.05 discarded the null hypothesis.

## 3. Results

Table 1 summarizes the demographics of the entire dataset. Table 2 summarizes the reasons for admission, with the most frequent being infections (43.58%), including respiratory infections and pneumonias, heart failure (29.14%), and delirium (22.94%).

The distribution of the frailty test outcomes is summarized in Table 3; 68.54% of the patients FFI scores were frail, and 97.68% were either frail or near frail, while 64.44% of the patients were at a nutritional risk (MNA-SF score < 8). However, most of the patients (88.61%) were mildly to highly independent according to BIS, and 80.68% were almost normal in the Pfeiffer’s scale of dementia.

Figure 2 and Figure 3 compare the survival probability Kaplan–Meier estimates of the prevalent and incident delirium cohorts, respectively, versus the non-delirium cohort at four censoring points. Log-rank test *p* values summarized in Table 4 show that delirium patients had a worse prognosis for survival at all the considered censoring times and cohorts.

Figure 4, Figure 5, Figure 6, Figure 7, Figure 8 and Figure 9 display, graphically, the results of the multivariate Cox’s regression at the 2-year censoring time, over the demographic, comorbidities, and pharmacological variables for the prevalent and incident delirium cohorts. The global log-rank significance of the regression coefficients of the demographic variables in Figure 4 and Figure 5 were highly significant (p<0.000001), contrary to the comorbidities in Figure 6 and Figure 7 that were not significant, while the pharmacological variables in Figure 8 and Figure 9 were mildly significant (p<0.05) and not significant (p<0.1), respectively.

Table 5 and Table 6 contain the variables with significant HR for the prevalent and incident cohorts, respectively, at considered censoring times. High HR variables identified in both delirium cohorts at some censoring times were age, gender (male), the number of drugs, the occurrence of readmissions before 30 days after discharge, FFI weight loss, SPPB time to sit and stand up, cholesterol (dyslipidemia), head trauma, cerebrovascular disease, and the use of diuretics was high. Specific HRs in the prevalent cohort were anticoagulated, quetiapine, and inhaled bronchodilators. Specific HRs in the incident cohort were α-adrenergic antagonist and 5α testosterone inhibitors, which were additional risk factors. Specifically, variables with the most significant 2-year hazard risks (HR(95%CI)) for patients with delirium were the following: male gender (0.43 (0.26,0.69)), weight loss (0.45 (0.26,0.74)), sit and stand up test (0.67 (0.49,0.92)), readmission in 30 days after discharge (0.50 (0.30,0.80)), cerebrovascular disease (0.45 (0.27,0.76)), head trauma (0.54 (0.29,0.98)), number of prescribed drugs (polypharmacy) (1.10 (1.03,1.18)), and the use of diuretics (0.57 (0.34,0.96)).

## 4. Discussion

The sample contained very old patients that were living in the community, in their own houses (71.52%), and some living alone (26.31%); thus, their basal (1 month before hospital admission) performance in the activities of daily living (ADL) was high, and most (89.00%) were either fully independent or had low-level dependency. However, at hospital admission, most patients were frail (69% scored FFI above three, 64% scored SPPB below three). Moreover, 17% had malnutrition and 47% were at risk of malnutrition, similar to [54]. The average Charlson comorbidity index of the sample was 6.39, which corresponded to less than a 2% 10-year survival probability [55]. Their admittance was due to infectious causes, exacerbations of heart failure, and delirium. Infectious causes for admittance are becoming prevalent in the ageing population, amounting to one-third of death causes in people older than 65 years [56].

Both frailty and delirium are associated with vulnerability and a lack of physiologic and cognitive reserve against insults [20,57]. Consequently, delirium HRs are strongly related to the frailty status of the subjects, with specific HRs being age, polypharmacy, hospital readmission, low scores in the sit and stand up test, weight loss, head trauma, cerebrovascular disease, dyslipidemia, and the use of diuretics. Dyslipidemia is a high risk factor [58] for cerebrovascular disease, which was ranked second among the main causes of death by the WHO in 2019 (https://www.who.int/es/news-room/fact-sheets/detail/the-top-10-causes-of-death, accessed on 1 February 2022).

A recent meta-analysis reported that the lower resistance in the sit and get up test, a smaller calf circumference, and weight loss are quite significant frailty risk factors [59]. Weight loss is often the last of the five characteristics of the Fried frailty phenotype to manifest, and it is extremely difficult to revert [60]; hence, it increases delirium mortality. The strength of the lower limbs measured by the sit and stand up test is a biomarker of disease severity [61] that is partly due to malnutrition and a lack of exercise [62]. These factors contribute to ageing adverse events, such as falls, which can produce head trauma that increases mortality in older patients with anticoagulant pharmacy [63], a known risk for older people [64]. Readmission before 30 days of discharge should be considered as another frailty indicator, but has not often been recognized as such.

Ageing is associated with physiological changes that affect drug pharmacokinetics and pharmacodynamics. This study found that increasing the number of drugs was a significant hazard risk for mortality, in agreement with recent findings [65,66,67,68,69,70,71,72,73,74,75]. Dehydration and acute kidney injury are well-know precipitating factors for delirium [1,2,3,9]; therefore, it is not surprising that diuretics were a highly significant hazard risk, because they are widely used as treatments of common diseases: arterial hypertension, heart failure, and renal disease. Electrolyte disturbances cause most side effects, such as delirium, raising concerns regarding diuretics overuse [76]. In the prevalent cohort, the study found inhaled bronchodilators, which have a known effect on the generation of heart arrhythmias [77], and quetiapine, which is an antipsychotic medicament used for patients whose neuropsychiatric symptoms have not responded to other alternative pharma. This finding was in agreement with literature reporting that quetiapine increases mortality by 2% [78]. However, recent publications propose low-dose quetiapine for delirium prevention in critically ill patients [79]. Here, its use could be interpreted as an indicator that delirium overlaps with dementia for some patients, as often found in the literature [14]. However, dementia was not found to be a significant hazard risk for mortality after delirium in this study. In the incident cohort, we found that urology indicated medicaments, such as α-adrenergic antagonist and 5α testosterone inhibitors, usually treating symptoms associated with benign prostatic hyperplasia. This finding was closely related with the higher mortality risk for males.

### Limitations

The identification of mortality risk factors associated with delirium in a cohort was heavily dependent on its demographics and prevalent conditions. Another limitation was that the cohort was recruited in a single site. We did not separately analyse the core symptoms of delirium that could provide additional insights [39]. It was not possible to evaluate recuperation from delirium after hospital discharge [12]. At hospital discharge, it was not feasible to repeat the physical and cognitive assessments due to discharge procedures. There was a lack of detailed records on the duration of the delirium, which could be a target for further research. The management of delirious patients is very delicate, so that the patient assessment and the realization of the physical and cognitive tests may lead to an unintended estimation bias in the evaluation of their hazard risks. The information extracted from the EHR, such as pharmacological data, should not produce any specific estimation bias.

## 5. Conclusions and Future Work

The objectives of the present study were two-fold: firstly, the assessment of an increased mortality due to delirium in an already frail population; secondly, the identification of high-risk variables. A comparison of the survival probability curves confirmed that delirium was a factor for greater mortality at the four censoring times considered. The findings of the study, in decreasing order of hazard risk significance, were the frailty indices, polypharmacy, the use of diuretics, and some comorbidities associated with delirium onset, such as high cholesterol, cerebrovascular disease, and head trauma.

## Figures and Tables

**Figure 1 ijerph-19-02247-f001:**
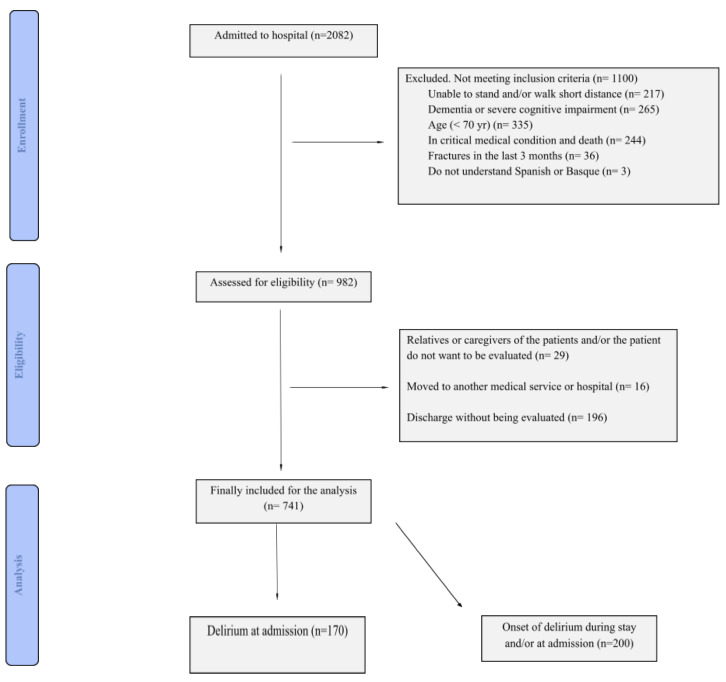
Flow diagram of the recruitment process.

**Figure 2 ijerph-19-02247-f002:**
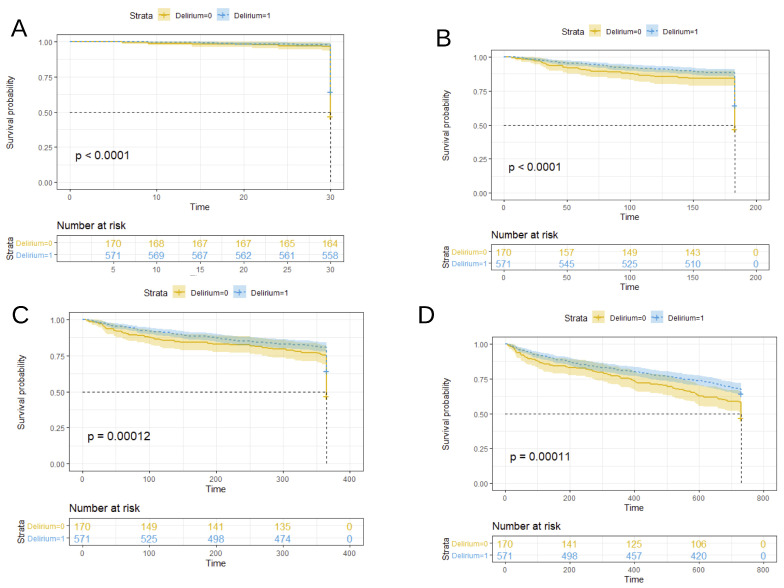
Kaplan–Meier survival curves for patients diagnosed with delirium at admittance (prevalent cohort) versus patients without delirium diagnosis. Data were censored after 1 month in plot (**A**), 6 months in plot (**B**), 1 year in plot (**C**), and 2 years in plot (**D**). Note: In the plots, delirium = 1 for patients with no delirium diagnosis (blue curve in the plots). Time was expressed in days. The *p*-value shown in the plots corresponds to the log-rank test comparison of no delirium versus delirium survival probability curves.

**Figure 3 ijerph-19-02247-f003:**
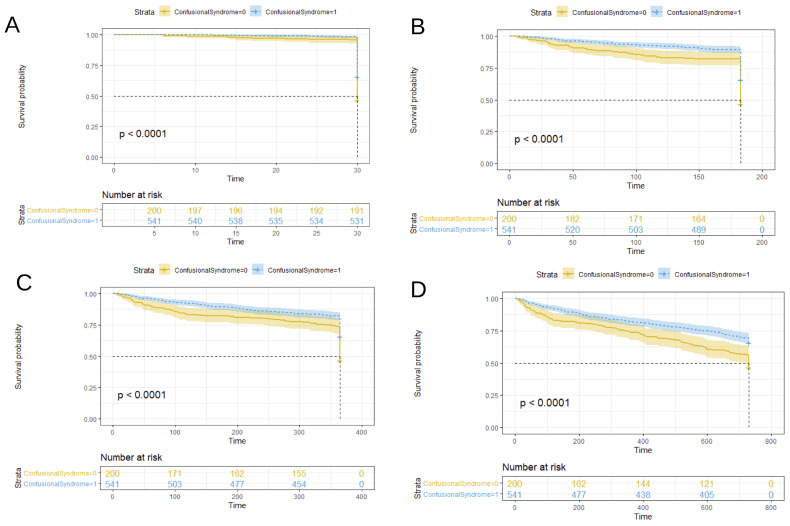
Kaplan–Meier survival curves for patients diagnosed with delirium at admittance or during the stay (incident cohort) versus patients without delirium diagnosis. Data were censored after 1 month in plot (**A**), 6 months in plot (**B**), 1 year in plot (**C**), and 2 years in plot (**D**). Note: In the plots, ConfusionalSyndrome = 1 for patients with no delirium diagnosis (blue curve in the plots). Time was expressed in days. The *p*-value shown in the plots corresponds to the log-rank test comparison of no delirium versus delirium survival probability curves.

**Figure 4 ijerph-19-02247-f004:**
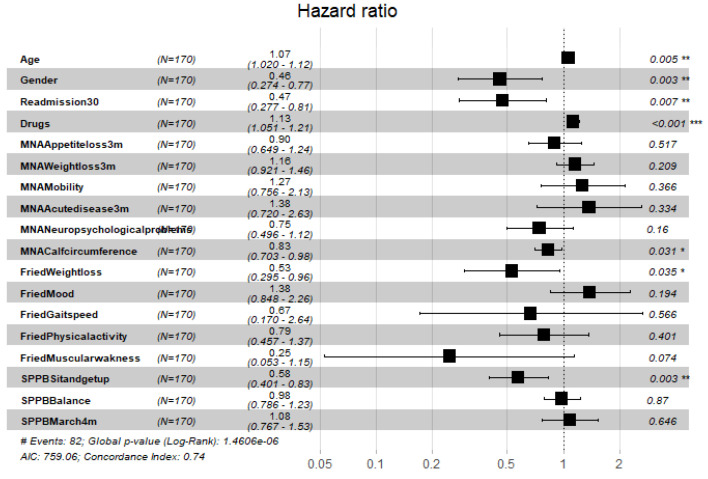
Hazard ratios of demographic variables and frailty scores for the prevalent cohort, data censored at 2 years. Note: left to right columns are the variable name, population size, HR (95% confidence interval), graphical representation of HR (95%CI), and-*p*-value $Pr\left(>\left|z\right|\right)$. AIC—Akaike Information Criterion; signif. codes: 0 ‘***’ 0.001 ‘**’ 0.01 ‘*’ 0.05 ‘.’ 0.1 ‘ ’ 1; concordance is the probability of agreement between two random observations; global score log-rank test was significant at 0.05.

**Figure 5 ijerph-19-02247-f005:**
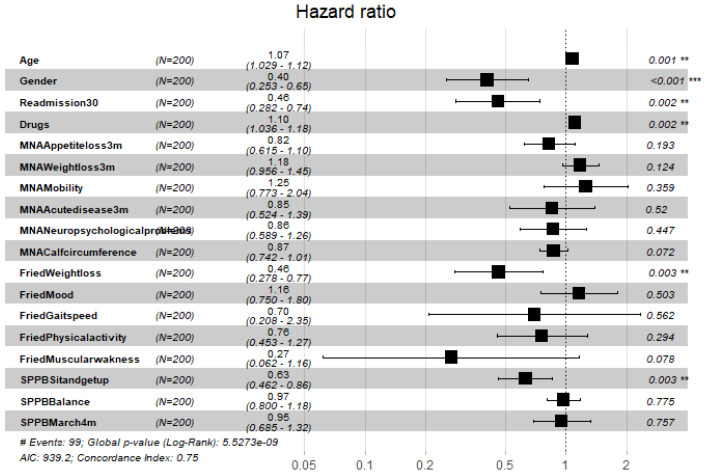
Hazard ratios of demographic variables and frailty scores for the incident cohort, data censored at 2 years. Note: left to right columns are the variable name, population size, HR (95% confidence interval), graphical representation of HR (95%CI), and *p*-value $Pr\left(>\left|z\right|\right)$. AIC—Akaike Information Criterion; signif. codes: 0 ‘***’ 0.001 ‘**’ 0.01 ‘*’ 0.05 ‘.’ 0.1 ‘ ’ 1; concordance is the probability of agreement between two random observations; global score log-rank test was significant at 0.05.

**Figure 6 ijerph-19-02247-f006:**
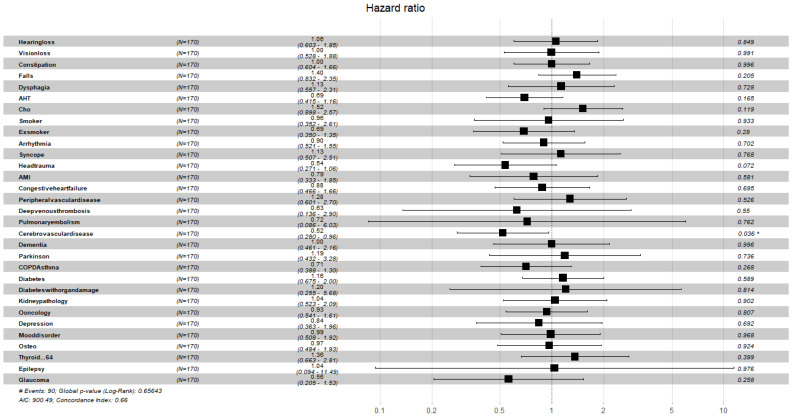
Hazard ratios of comorbidities for the prevalent cohort, data censored at 2 years. Note: left to right columns are the variable name, population size, HR (95% confidence interval), graphical representation of HR (95%CI), and *p*-value $Pr\left(>\left|z\right|\right)$. AIC—Akaike Information Criterion; signif. codes: 0 ‘***’ 0.001‘**’ 0.01 ‘*’ 0.05 ‘.’ 0.1 ‘ ’ 1; concordance is the probability of agreement between two random observations; global score log-rank test was significant at 0.05.

**Figure 7 ijerph-19-02247-f007:**
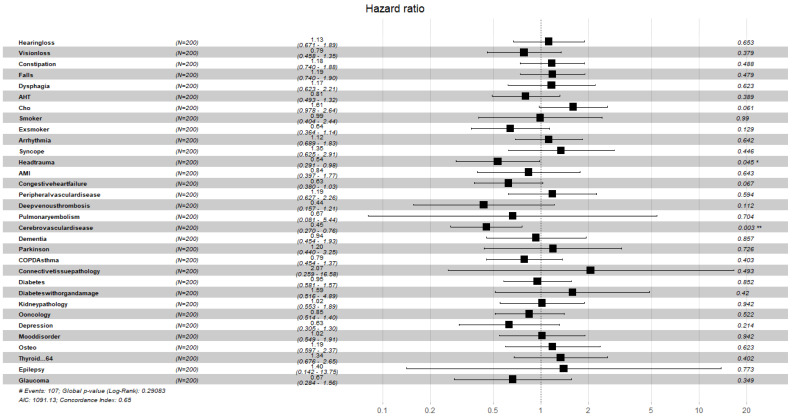
Hazard ratios of comorbidities for the incident cohort, data censored at 2 years. Note: left to right columns are the variable name, population size, HR (95% confidence interval), graphical representation of HR (95%CI), and *p*-value $Pr\left(>\left|z\right|\right)$. AIC—Akaike Information Criterion; signif. codes: 0 ‘***’ 0.001 ‘**’ 0.01 ‘*’ 0.05 ‘.’ 0.1 ‘ ’ 1; concordance is the probability of agreement between two random observations; global score log-rank test was significant at 0.05.

**Figure 8 ijerph-19-02247-f008:**
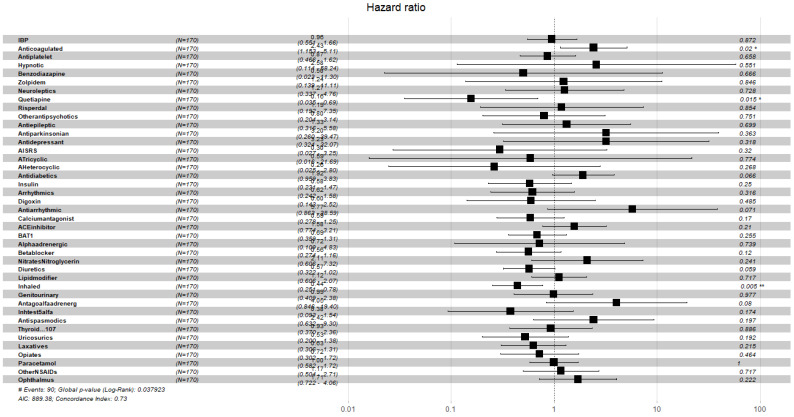
Hazard ratios of pharmacological variables for the prevalent cohort, data censored at 2 years. Note: left to right columns are the variable name, population size, HR (95% confidence interval), graphical representation of HR (95%CI), and *p*-value $Pr\left(>\left|z\right|\right)$. AIC—Akaike Information Criterion; signif. codes: 0 ‘***’ 0.001 ‘**’ 0.01 ‘*’ 0.05 ‘.’ 0.1 ‘ ’ 1; concordance is the probability of agreement between two random observations; global score log-rank test was significant at 0.05.

**Figure 9 ijerph-19-02247-f009:**
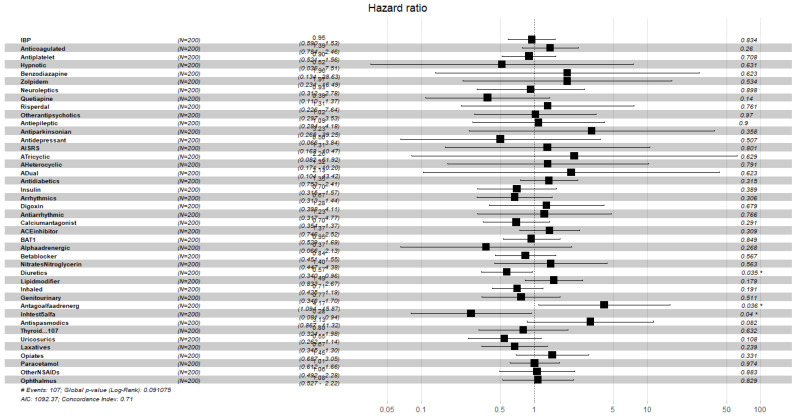
Hazard ratios of pharmacological variables for the incident cohort, data censored at 2 years. Note: left to right columns are the variable name, population size, HR (95% confidence interval), graphical representation of HR (95%CI), and *p*-value $Pr\left(>\left|z\right|\right)$. AIC—Akaike Information Criterion; signif. codes: 0 ‘***’ 0.001 ‘**’ 0.01 ‘*’ 0.05 ‘.’ 0.1 ‘ ’ 1; concordance is the probability of agreement between two random observations; global score log-rank test was significant at 0.05.

**Table 1 ijerph-19-02247-t001:** Demographics information of the cohorts.

Variable	Categories	N Total	%	ND (N = 541)	P (N = 170)		I (N = 200)	
				**%**	**%**	p	**%**	p
Gender	Male	382	51.55	53.97	43.53	0.017	45.00	0.030
	Female	359	48.45	46.03	56.47		55.00	
Weight (kg)	Mean (SD)	67.43 (13.60)		68.02 (13.00)	67.43 (13.61)	<0.001	67.43 (13.60)	<0.001
Age (years)	Mean (SD)	84.37 (6.76)		83.43 (6.67)	84.37 (6.76)	<0.001	84.37 (6.76)	<0.001
MS	Married	265	35.60	36.41	30.58	0.109	33.00	0.543
	Single	65	8.90	9.06	8.24	0.778	8.00	0.652
	Divorced	13	1.80	1.48	2.35	0.499	2.50	0.348
	Widowed	265	35.70	31.98	47.65	<0.001	47.00	<0.001
	NA	133	18.00	21.07	11.18	0.009	9.50	<0.001
HD	Yes	501	67.61	62.47	79.41	0.002	81.50	<0.001
	No	241	32.39	37.53	20.59		18.50	
NWS	Yes	419	56.54	52.12	66.47	0.443	68.00	0.189
	No	322	43.46	47.88	33.53		32.00	
Living at	Own home	530	71.52	70.42	69.41	<0.001	70.50	<0.001
	Alone	195	26.31	25.69	25.88	0.454	24.00	0.120
	Other’s home	62	8.36	6.28	11.18	0.248	13.00	0.018
	Retirement house	64	8.63	6.09	16.47	<0.001	14.50	0.003
Polypharmacy	Oligopharma <5	178	24.02	25.32	22.35	0.562	20.50	0.173
	Moderate (5–9)	358	48.31	49.72	44.71	0.267	45.00	0.254
	Severe (>9)	205	27.67	24.96	32.94	0.072	34.50	0.010
Prevalent delirium		170	22.94					
Incident delirium		200	26.99					

Note: ND—no delirium; P—prevalent cohort; I—incident cohort (prevalent + delirium while in the hospital); for dichotomic variables, *p*-value corresponds toWelch’s *t*-test, for categorical variables, *p*-value corresponds to ANOVA; SD—standard deviation; MS—marital status. Prevalent delirium = admittance reason was delirium; Incident delirium = admittance reason was delirium or there was diagnostic of delirium during stay. HD—has descendancy? NWS—needs walking stick.

**Table 2 ijerph-19-02247-t002:** Reasons for admission.

Reason for Admission	N	%
Heart failure	216	29.14
Infection	323	43.58
Anemia	79	10.66
CAL	101	13.63
DM	48	6.47
Delirium	170	22.94
Fall	119	16.05
Neurological	67	9.04
Oncology	35	4.72
OC	63	8.50
Kidney failure	96	12.95

Note: CAL—chronic airflow limitation; DM—diabetes mellitus; OC—cardiopathies different from heart failure.

**Table 3 ijerph-19-02247-t003:** Distribution of the frailty and cognitive test scores.

Test	Score	N	%
SPPB	Minimum (10–12)	95	12.91
	Light (7–9)	169	22.97
	Moderate (4–6)	232	31.52
	Severe (0–3)	240	32.60
FFI	Robust (0)	17	2.32
	Pre-frail (1–3)	213	29.14
	Frail (>3)	501	68.54
BIS	Independent	237	35.06
	Mild (>60)	362	53.55
	Moderate (40–55)	57	8.43
	Severe (20–35)	13	1.92
	Total Dependence (<20)	7	1.04
MNA-SF	Normal (12–14)	261	35.56
	At Risk (8–11)	348	47.41
	Poor nutrition (0–7)	125	17.03
PBSTD	Normal (0–2)	465	63.27
	Light (3–4)	128	17.41
	Moderate (5–7)	108	14.70
	Severe (8–10)	34	4.62

Note: SPPB—Short Physical Performance Battery; FFI—Fried’s frailty index; BIS—Barthel’s index score; PBSTD—Pfeiffer’s Brief Screening Test for Dementia.

**Table 4 ijerph-19-02247-t004:** Log-rank test results of comparison of survival probability curves of delirium cohorts versus no delirium cohort at chosen censoring dates.

Cohort	1 month	6 month	1 year	2 years
prevalent	$p=5\times 10^{-5}$ ***	$p=7\times 10^{-5}$ ***	$p=1\times 10^{-4}$ ***	$p=1\times 10^{-4}$ ***
incident	$p=1\times 10^{-6}$ ***	$p=1\times 10^{-6}$ ***	$p=3\times 10^{-6}$ ***	$p=2\times 10^{-6}$ ***

Note: prevalent—delirium at admission (N = 170); incident—delirium during stay (N = 200). ***—strongly significant *p* < 0.001.

**Table 5 ijerph-19-02247-t005:** Summary of variables with significant hazard risk found by Cox’s regression carried out over families of variables for the prevalent cohort, data censored after 1 month, 6 months, 1 year, and 2 years.

	1 month	6 months	1 year	2 years
**Variable**	**HR** **(95%CI)**	**HR** **(95%CI)**	**HR** **(95%CI)**	**HR** **(95%CI)**
Age	1.05 (1.00,1.09)	1.04 (1.00,1.09)	1.05 (1.00,1.10)	1.05 (1.00,1.10)
Gender	0.63 (0.38,1.02)	0.52 (0.31,0.86)	0.48 (0.28,0.80)	0.48 (0.28,0.80)
R30	0.65 (0.38,1.11)	0.53 (0.31,0.90)	0.53 (0.30,0.89)	0.52 (0.30,0.89)
ND	1.08 (1.01,1.15)	1.10 (1.03,1.18)	1.13 (1.05,1.21)	1.13 (1.05,1.21)
MNA-SF-CC	0.90 (0.76,1.06)	0.85 (0.72,1.00)	0.83 (0.70,0.98)	0.83 (0.70,0.98)
FFI-WL	0.55 (0.30,0.99)	0.59 (0.32,1.05)	0.50 (0.27,0.90)	0.53 (0.29,0.94)
SPPB-SUG	0.72 (0.504,01.02)	0.68 (0.47,0.98)	0.65 (0.45,0.94)	0.64 (0.44,0.92)
Falls	1.65 (0.98,2.74)	1.53 (0.91,2.56)	1.46 (0.87,2.46)	1.40 (0.83,2.35)
Cholesterol	1.55 (0.91,2.61)	1.61 (0.95,2.72)	1.63 (0.96,2.76)	1.52 (0.89,2.57)
Head Trauma	0.51 (0.262,1.00)	0.54 (0.28,1.07)	0.53 (0.26,1.06)	0.54 (0.27,1.06)
CD	0.49 (0.27,0.87)	0.47 (0.26,0.84)	0.48 (0.26,0.87)	0.52 (0.28,0.96)
Anticoagulated	1.74 (0.85,3.54)	2.11 (1.01,4.38)	2.37 (1.12,4.98)	2.43 (1.15,5.11)
Quetiapine	0.23 (0.05,0.93)	0.16 (0.03,0.67)	0.15 (0.03,0.66)	0.16 (0.03,0.69)
Diuretics	0.61 (0.34,1.08)	0.58 (0.32,1.03)	0.60 (0.33,1.06)	0.57 (0.32,1.02)
IB	0.51 (0.29,0.87)	0.43 (0.24,0.75)	0.43 (0.24,0.75)	0.44 (0.25,0.78)

Note: HR(95% CI)—mean hazard risk (95% confidence interval). HR significant = *p* < 0.05. Nonsignificant variables were excluded from the table. Prevalent cohort: patients admitted with delirium (N = 170); gender—0 if male; R30—0 if patient experienced readmission in the 30 days after discharge; ND—number of drugs in polypharmacy; FFI—Fried Frailty Index; FFI-WL—weight loss from the FFI test; SPPB-SUG—sit and stand up from the SPPB test; MNA-CC—calf circumference from the MNA-SF evaluation. Falls—number of fall events in the previous month; Head Trauma—0 if there was head trauma at admission; CD—0 if there was history of cerebrovascular disease; Anticoagulated—use of any anticoagulation drug; quetiapine = 0 if there was use of quetiapine; Diuretics—use of any diuretic drug; IB = 0 if there was use of inhaled bronchodilators.

**Table 6 ijerph-19-02247-t006:** Summary of variables with significant hazard risk found by Cox’s regression carried out over families of variables for the incident cohort, data censored after 1 month, 6 months, 1 year, and 2 years.

	1 month	6 months	1 year	2 years
**Variable**	**HR** **(95%CI)**	**HR** **(95%CI)**	**HR** **(95%CI)**	**HR** **(95%CI)**
Age	1.05 (1.00,1.1)	1.05 (1.01,1.10)	1.06 (1.01,1.11)	1.07 (1.02,1.11)
Gender	0.65 (0.42,1.0)	0.48 (0.30,0.75)	0.43 (0.26,0.69)	0.43 (0.26,0.69)
R30	0.62 (0.39,1.0)	0.51 (0.32,0.83)	0.51 (0.31,0.81)	0.50 (0.30,0.80)
ND	1.07 (1.00,1.14)	1.08 (1.02,1.15)	1.10 (1.03,1.17)	1.10 (1.03,1.18)
FFI	1.72 (0.92,3.2)	1.82 (0.98,3.41)	1.83 (0.98,3.41)	1.81 (0.97,3.36)
FFI-WL	0.50 (0.30,0.83)	0.51 (0.31,0.84)	0.42 (0.25,0.70)	0.45 (0.26,0.74)
SPPB-SGU	0.74 (0.55,1.01)	0.71 (0.52,0.97)	0.68 (0.50,0.93)	0.67 (0.49,0.92)
Cholesterol	1.62 (0.99,2.64)	1.68 (1.03,2.76)	1.70 (1.03,2.80)	1.61 (0.97,2.64)
Head Trauma	0.54 (0.29,0.99)	0.56 (0.30,1.03)	0.56 (0.30,1.04)	0.54 (0.29,0.98)
CD	0.47 (0.29,0.77)	0.44 (0.26,0.72)	0.44 (0.26,0.73)	0.45 (0.27,0.76)
Diuretics	0.58 (0.35,0.97)	0.59 (0.34,0.98)	0.59 (0.35,0.99)	0.57 (0.34,0.96)
α-Adrenergic Antagonist	2.68 (0.74,9.65)	3.69 (0.98,13.78)	3.92 (1.07,14.37)	4.17 (1.09,15.87)
5α Testosterone inhibitors	0.36 (0.11,1.18)	0.31 (0.09,1.04)	0.28 (0.08,0.94)	0.28 (0.08,0.94)

Note: HR(95% CI)—mean hazard risk (95% confidence interval). HR significant when *p* < 0.05. Nonsignificant variables were excluded from the table. Incident cohort: patients admitted with delirium or that experienced delirium during hospital stay (N = 200). Gender—0 if male; R30—0 if patient experienced readmission in the 30 days after discharge; ND = number of drugs in polypharmacy; FFI—Fried Frailty Index; FFI-WL—weight loss from the FFI test; SPPB-SUG—sit and stand up from the SPPB test. Head Trauma—0 if there was head trauma at admission; CD—0 if there was history of cerebrovascular disease; Diuretics—use of any diuretic drug; α-Adrenergic Antagonist—number of α-adrenergic antagonist drugs; 5α Testosterone inhibitors—0 use of 5α testosterone inhibitors.

## Data Availability

Upon acceptance, the data will be published as .csv and .xlx files on zenodo.org.

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
