# Peer review of "Survival of Frail Elderly with Delirium"

_ijerph, 2022, doi:10.3390/ijerph19042247_

Round 1

Reviewer 1 Report

I feel that delirium is a very important geriatric syndrome, especially in the hospital setting where many preventive actions could be taken.
But I feel that this manuscript has many shortcomings.
The introduction is poorly written.
The methods are disorganised, and there is a lack of information on the variables. In the results there are many variables that are not described.
The results are not well thought out.
Furthermore, I have an important doubt about the assessment of the physical tests of the elderly during hospital admission.

I have attached the manuscript with comments.

Author Response

Dear reviewer, thanks for your time and the suggestions for improvement of the paper

Our responses are in the attached document

Reviewer 2 Report

The Authors, have presented an interesting manuscript. However, I have comments to the article:

In section Results, Authors have presented the same information as in section Matreials and Methods, it should be checked and modified.  

In section Disscussion there are information about tables – remove the reference to tables from Disscussion.

In section “Conclusions and future work” should be implications of your research and practical implications for future, it should be added.

Author Response

Dear reviewer, thanks for your time and the suggestions for improvement of the paper. Our responses are in the attached document

Reviewer 3 Report

Thanks to the authors for their effort and work. It is certainly a topic of interest and brings a significantly important insight into the clinical context.
However, before making a decision on its publication, I must recommend a number of improvements:

1.- Introduction: Beyond the conceptualisation, I would appreciate a background study on the research topic as the theoretical part of the manuscript is too short and decontextualised.

2.- Method and Results: Both sections are very well developed following the appropriate inclusion and exclusion protocols as well as a statistical analysis appropriate to the type of study proposed.

3.- Discussion and conclusions: The extension of the introductory section will surely yield new bibliography that will help to clarify these sections. They are well developed and consistent but, surely, an extension of the bibliography would help to qualify the results and conclusions obtained.

Finally, it should be pointed out that this is a well-prepared and well-designed study. Only small actions need to be taken regarding the expansion of the theoretical section and the expansion of the conclusions. 

Best regards.

Author Response

(The authors gave the same response as above.)

Round 2

Reviewer 1 Report

The authors have significantly improved the manuscript, it is evident that it is much more elaborate, but I feel it still has important shortcomings from the beginning of the methodology used, which are difficult to resolve now:
It is not clear when and how the assessments were made and it seems rather complicated to obtain all these data in a systematic way.
Delirium has a hospital average remission time of about 6 days, a patient may remit in 1 day, or it may take 12 days, or after remission there may not be an immediate correct cognitive function. 
This is the main limitation of this study and anyone who works in a hospital environment or with the older people knows how difficult it is to obtain these data.

The authors also continue to make errors in the format of a scientific manuscript:
Table and figure legends do not go in the title.
In English, decimals are followed by full stops.
The acronym % does not need to be followed by each number if the column is all percentages.
Elder is a word in disuse, being better to use the word older.

The results thus obtained have a very, very important bias and the conclusions should also be worded differently.
